# Role of Microbiota-Derived Extracellular Vesicles in Gut-Brain Communication

**DOI:** 10.3390/ijms22084235

**Published:** 2021-04-19

**Authors:** Carlos M. Cuesta, Consuelo Guerri, Juan Ureña, María Pascual

**Affiliations:** 1Department of Molecular and Cellular Pathology of Alcohol, Príncipe Felipe Research Center, 46012 Valencia, Spain; cmcuesta@cipf.es (C.M.C.); cguerri@cipf.es (C.G.); jurena@cipf.es (J.U.); 2Department of Physiology, School of Medicine and Dentistry, University of Valencia, Avda. Blasco Ibáñez, 15, 46010 Valencia, Spain

**Keywords:** microbiota, bacteria, extracellular vesicles, brain, neuropathology

## Abstract

Human intestinal microbiota comprise of a dynamic population of bacterial species and other microorganisms with the capacity to interact with the rest of the organism and strongly influence the host during homeostasis and disease. Commensal and pathogenic bacteria coexist in homeostasis with the intestinal epithelium and the gastrointestinal tract’s immune system, or GALT (gut-associated lymphoid tissue), of the host. However, a disruption to this homeostasis or dysbiosis by different factors (e.g., stress, diet, use of antibiotics, age, inflammatory processes) can cause brain dysfunction given the communication between the gut and brain. Recently, extracellular vesicles (EVs) derived from bacteria have emerged as possible carriers in gut-brain communication through the interaction of their vesicle components with immune receptors, which lead to neuroinflammatory immune response activation. This review discusses the critical role of bacterial EVs from the gut in the neuropathology of brain dysfunctions by modulating the immune response. These vesicles, which contain harmful bacterial EV contents such as lipopolysaccharide (LPS), peptidoglycans, toxins and nucleic acids, are capable of crossing tissue barriers including the blood-brain barrier and interacting with the immune receptors of glial cells (e.g., Toll-like receptors) to lead to the production of cytokines and inflammatory mediators, which can cause brain impairment and behavioral dysfunctions.

## 1. Introduction

The human gastrointestinal tract contains a diverse range of bacterial species called microbiota that play a key role in the body homeostasis. This complex ecosystem is involved in the immune system, energy efficiency, the synthesis of essential components or even the emotional state. Increasing evidence has associated gut microbiota with both gastrointestinal and extra-gastrointestinal diseases such as the central nervous system (CNS) in which bidirectional communication between the CNS and gut microbiota or the gut-brain axis has been of special interest in recent years. Under healthy physiological conditions, beneficial bacteria can protect against different diseases such as asthma, food allergy and type 1 diabetes [1]. However, dysfunctions in gut microbiota or dysbiosis affect host immunity and are not only associated with chronic disease (obesity, diabetes, cardiovascular disease, etc.) [2] but also with mental disorders, brain dysfunction or behavioral impairments such as Alzheimer’s or Parkinson’s diseases [3,4,5,6].

Recent studies have demonstrated participation in gut-brain communication on different pathways such as the autonomic nervous system, endocrine routes, the hypothalamic-pituitary-adrenal axis and the immune system [7]. However, new evidence reveals the role of extracellular vesicles (EVs) from microbiota species including bacteria, which can modulate the immune and neuroimmune response of host cells [1]. Likewise, several pieces of evidence indicate that multiple pathways are involved in the communication from the human intestinal microbiota to the CNS including vagal afferent nerves, immune and hypothalamic-pituitary-adrenal (HPA) axis modulation and the production of active metabolic products [8].

In this review, we focus on the role of EVs from gastrointestinal microbiota in modulating the immune response during neurodevelopmental processes of brain dysfunctions. We also review how bacterial EVs are capable of crossing intestinal and brain barriers and their multiple molecules (e.g., lipoproteins, lipopeptides, lipopolysaccharide (LPS) or DNAs), to activate immune signaling cascades by engaging Toll- or NOD-like receptors (TLRs or NLRs) in glial cells, which leads to a neuroinflammatory immune response that can trigger brain and behavior dysfunctions.

## 2. Gut Microbiota

The intestinal microbiota is made up of 100 trillion cells [9], more than the tissue cells in the entire body, of which the vast majority are bacterial species with a minority of fungi and archaea kingdoms and even a gut virome dominated by bacteriophages that replicates in gut bacteria [10]. A healthy microbiome functions as any other organ in the human body and is a complex ecosystem where hundreds of species exist with one another and with human cells. Their functions can be classified as metabolic, by increasing the absorption of nutrients and the extraction of energy and by providing new biochemical pathways and metabolites for the body and protective, by participating in the development of innate and adaptive immunity and preventing the intestine from being invaded by pathogenic species that would occupy the microbiota niche. Finally, the microbiome can also play a structural role by regulating intestinal cells, their intercellular junctions and mucus production [11].

The healthy human microbiota performs similar functions on taxa and metabolic pathways in all humans and even the existence of a common bacterial core has been hypothesized. Some microbial species are present in all human individuals [12]. However, there are no two microbiotas with identical bacterial species and abundances and are as unique as fingerprints. Diet is the main factor that modifies the microbiota [13]. Nevertheless, other factors such as the use of antibiotics or probiotics, age, stress, host genotype or even fetus delivery can benefit a particular bacterial community [14,15].

Certainly, microbiota homeostasis can be broken, whose term is dysbiosis, which is characterized by the loss of beneficial microorganisms, the appearance or increase of harmful microorganisms or the loss of overall microbial diversity [16]. Dysbiosis leads to several disorders in which the activation of the immune system and inflammatory response are usually common mechanisms in them all (e.g., Crohn’s disease, ulcerative colitis, type II diabetes, cardiovascular diseases [17] and even neuropsychiatric disorders [18]).

The inner intestine surface is made up of a single layer of cells with a permeable outer mucosa that allows the exchange of nutrients, which makes the intestine a vulnerable tissue against pathogens. The gastrointestinal tract’s immune system is composed of GALT (gut-associated lymphoid tissue), the most important structural component of mucosa-associated lymphoid tissue, with Peyer’s patches, which are lymphoid follicles located in the mucosa that play an important role in not only removing pathogens in the gut but also in maintaining a balance with the commensal and pathogenic microbiotas in the gut [19]. Commensal and pathogenic bacteria acquire different “interests” and outcomes in their biological interaction with the host. Pathogenic enteric microorganisms perform an infectious process, which usually begins with the capacity to adhere to the brush border of intestinal cells and to disrupt the epithelium [20] and also contain “pathogenicity islands”, which are specific regions in their chromosomes that enclose virulence genes [21]. Both commensal and pathogenic bacteria interact with the immune system but commensal microorganisms are necessary to modulate homeostasis [22] and even collaborate with the tissue [23], while the pathogenic microbiota disturbs the immune system to parasitize its host [24].

## 3. Gut Microbiota and Brain Communication

The link between the gastrointestinal and nervous systems has been theorized since Ancient Greece but the gut-brain axis term was postulated by Ivan Pavlov in 1902, who demonstrated a cephalic phase that interacted with both gastric and pancreatic secretions [25]. In the last few decades, the microbiota-gut-brain axis term has emerged with discoveries made from using germ-free (GF) animals. Brain biochemistry and behavior are similar to normal mice in young GF animals treated with microbiota [26] and several probiotics such as *Bifidobacterium infantis* can exert beneficial therapeutic effects on mental disorders [27]. Conversely, the administration of antibiotics is able to induce mental disorders in patients [28] and treatment with specific bacterial strains can cause alterations to behavior and the nervous system with no immune response activation [29].

One of the main pathways of potential communication between the gut microbiota and the brain is the autonomic nervous system, which includes the enteric nervous system and the vagus nerve. This nerve in intestines regulates the contraction of smooth muscles and glandular secretion and takes sensory information to the brain by playing a role in psychiatric disorders [30]. Bacteria might contact vagal signaling via vagal mechanoreceptors and chemoreceptors by altering contractile activity and producing substances such as short-chain fatty acids or cholecystokinin, respectively [31]. For instance, *Lactobacillus rhamnosus* is a probiotic bacterium that modulates the GABAergic system with beneficial effects on depression and anxiety behavior by using the vagus nerve as the main modulatory pathway between it and the CNS [32]. Indeed, a fecal microbiota transplant alleviated sepsis-associated encephalopathy through the vagus nerve [33]. The enteric nervous system can mediate gastrointestinal behavior independently of the CNS but both systems are communicated and neuronal changes in one of them can be manifested in the other [34]. Furthermore, gut microbiota can themselves release metabolites including neurotransmitters such as serotonin, dopamine, noradrenaline and GABA. These neurotransmitters and their precursors, hormone-like metabolites and short-chain fatty acids can travel through the circulation to the brain where they can act on respective receptors modulating neuronal and microglial functions [35]. In this gut-brain communication, endocrine pathways also participate such as enteroendocrine cells and the hypothalamic-pituitary-adrenal axis. Enteroendocrine cells are specialized epithelial cells with chemoreceptors (e.g., G protein-coupled receptors or TLRs) that respond to intestinal bacterial metabolites [36]. They are also capable of releasing products that act as intermediary compounds or messengers between the interior of the gastrointestinal tract and intrinsic primary afferent and vagal afferent neurons [37]. The hypothalamic-pituitary-adrenal axis is the main anatomical structure that mediates the stress response through the release of the corticotropin-releasing factor (CRF), which acts on the pituitary to secrete a corticotropic hormone (ACTH) and finally stimulates glucocorticoids synthesis and secretion from the zona fasciculata of the adrenal cortex [38]. This activation is controlled by neuronal and endocrine systems and glucocorticoids acquire negative feedback by inhibiting this axis, which leads to several harmful gastrointestinal disorders [39]. However, the influence of the gut microbiota on stress-induced hypothalamic-pituitary-adrenal axis alterations depends on the species and strain of the affected bacteria (e.g., anti-stress or pro-stress status) [40].

Finally, the last pathway with a strong interaction between the gut microbiota and the brain is the immune system. This communication is performed by cytokines (e.g., interleukin-22 (IL-22), IL-17 or IL-10) [41] through afferent nerves (e.g., the vagus nerve), the HPA axis or by crossing the blood-brain barrier (BBB) [42], which can alter CNS functions and lead to behavioral impairments [43]. However, recent evidence has demonstrated the role of extracellular vesicles released by bacteria as communicators between the gut microbiota and the brain and their participation in brain diseases [18].

## 4. Role of Extracellular Vesicles in the Host

One of the characteristics conserved in all organisms is the ability to secrete several types of membrane vesicles or extracellular vesicles (EVs). When EVs were discovered, their function was initially thought to be the removal of unnecessary compounds from cells [44]. However, this is only one characteristic among others because the most interesting functions are intercellular communication and exchanging proteins, lipids and nucleic acids with their respective target cell [45]. Among the most important components of the secretome in eukaryote cells are exosomes, which are membrane-derived microvesicles (30–150 nm) that contain proteins (e.g., tetraspanins, receptors ligands or adhesion molecules), RNAs and lipids (Figure 1). Under healthy physiological conditions, the functions of EVs can be summarized as intercellular communication, angiogenesis, cell survival, inflammation and immune response, coagulation and waste management [46,47,48,49]. 

These microvesicles might contain inflammatory molecules such as pro-inflammatory cytokines, TLR4, inflammation-associated RNAs and miRNAs, which contribute to the pathogenesis of different disorders such as cancer and inflammatory, autoimmune or neurodegenerative diseases, among others [50]. For instance, the involvement of exosomes in neuroinflammation has also been associated with mental disorders (e.g., depression, anxiety, bipolar disorder, schizophrenia) by altering the release of microglial exosomes [51] and after a toxic stimulus (e.g., ethanol) by increasing the release of astrocyte-derived EVs and their content in inflammation-related proteins [51]. As EVs can cross the BBB, these vesicles can be used as carriers of neuroinflammation by spreading the immune response. Indeed the content of inflammatory miRNAs in plasma EVs from human and mice with alcohol abuse is similar to their target genes in brain EVs [52].

## 5. Microbiota-Derived EVs

### 5.1. Biogenesis of Bacteria EVs

As in other organisms, bacteria can also produce EVs but their biogenesis and structure differ from the EVs released by eukaryotic cells (see Table 1 and Figure 1). As the vesicle content is similar to the intracellular content of cells from which vesicles derive (are secreted), before mentioning the biogenesis of bacterial EVs, we analyze the structure of Gram-positive and Gram-negative bacteria. Gram-positive bacteria have only one lipid bilayer with an extensive peptidoglycan polymer that protects the surface while Gram-negative bacteria have one inner and one outer membrane with a periplasmic space between these membranes and a peptidoglycan polymer linked with the outer membrane in the interior and an LPS linked with the surface [53].

All Gram-negative bacteria can produce outer membrane vesicles (OMVs), which are spherical and bilayered vesicles that derive from the outer membrane through bulging and pinching it off [54]. These vesicles are constantly released from the surface during bacterial growth and increase under specific situations such as stress conditions. They have a similar content to “mother” bacteria including outer membrane proteins, LPS, phospholipids and periplasmic constituents [55] (Figure 1). The vesicle biogenesis of these Gram-negative bacteria is categorized into three non-exclusive models in which host-associated factors can participate. One of them occurs when lipid asymmetry maintenance is compromised, which affects the lipid interactions between the components of the outer membrane and peptidoglycans by increasing the release of membrane vesicles. Extracytoplasmic stress responses such as the accumulation of misfolded or unfolded proteins on the outer membrane can also affect the release of membrane vesicles. The last model occurs by modifications in the interactions between LPS molecules induced by intrinsic and extrinsic (e.g., cationic antimicrobial peptides and cations) factors [56]. In addition, some Gram-negative bacteria also produce another type of vesicle at a low rate, which represents a new model of vesiculation with a double layer, termed “outer-inner membrane vesicles” (O-IMVs). These vesicles contain similar material to the cytoplasm and, during the biogenesis process, would involve momentary ruptures of the cell wall and plasma membrane, which may compromise cell viability. Particular importance is attached to these O-IMVs in the case of pathogenic bacteria in which EVs are associated with the transfer of DNA, toxins and other virulence factors [57] (Figure 1).

**Table 1 ijms-22-04235-t001:** Differences and similarities in EVs deriving from eukaryotic cells and bacteria.

Eukaryotic Organism	Bacteria
Spherical particles with a size range from 30–100 nm (exosomes), 100–1000 nm (MVs) or 500–2000 nm (apoptotic bodies). Stable at 37 °C for 24 h. Sensitive to high temperature [58] but stable in the frozen and freeze-dried states.	Spherical particles with a size range from 10–400 nm. The maximum size is smaller than eukaryotic EVs due to smaller sized bacterial cells. Stable at 37 °C for 24 h. Greater tolerance to hot temperatures [58]. Stable in the frozen and freeze-dried states.
Exosomes are commonly enriched in endosome-associated proteins.	Mainly composed of proteins and phospholipids of the outer membrane.
Exosomes and MVs are released by healthy and damaged cells. Apoptotic bodies are released by dying cells on an apoptotic pathway.	All Gram-negative bacteria produce outer membrane vesicles (OMVs) and possibly also all Gram-positive bacteria. Gram-negative bacteria can produce specific vesicles with a double layer using both the outer and inner membranes.
Originates in the plasma membrane except exosomes, which are made by the endocytic pathway.	Bacteria Gram-negative and Gram-positive have a different mechanism of vesicle formation due to their distinct membrane structure, which originates in the membrane.
They are released from cells by a variety of mechanisms depending on their mode of biogenesis and they are not released homogeneously by the membrane.	Production is not uniformly distributed along the bacteria surface but there are “hot spots”.
High heterogeneity in the composition of the surface and the interior.	High heterogeneity in the composition of the surface and the interior.
There are universal markers such as CD40 for microvesicles or flotillin for exosomes.	There are no universal markers for their identification due to their diversity.
EVs can contain different RNAs such as miRNA or mRNA but it is unusual for them to carry DNA.	EVs can contain genetic material and participate in horizontal gene transfer.
Harmful cells such as tumor cells present EVs with specific and useful contents for their survival.	In pathogenic bacteria, specific molecules have been found such as adhesins, toxins and/or immunomodulatory compounds as cargo of OMVs.
The main function is intercellular communication, except for apoptotic bodies, which facilitates phagocytosis.	They are more relevant as a mechanism to carry away toxic compounds for bacteria than in eukaryotic cells.
Production depends on the cell type and its physiology state.	Their production increases as a response to environmental stress.
A non-spontaneous biological process.	A non-spontaneous biological process.

Until recently, it was thought that Gram-positive bacteria could not produce EVs because it would be impossible for them to traverse the thick walls found in these organisms. However, vesicles have been recently isolated and analyzed from Gram-positive bacteria. These vesicles contain lipid bilayer-enclosed structures that are morphologically similar to Gram-negative bacteria. Nevertheless, Gram-positive EVs can still be distinguished from OMVs because they lack LPS and encapsulate periplasmic components (e.g., peptidoglycan and proteins) (Figure 1). Although the mechanism to release them is still an unsolved question, three non-mutually exclusive hypotheses have been postulated: they are forced through the wall by turgor pressure, enzymes with cell wall-modifying properties are released with EVs to increase pore size and permeability to facilitate the delivery and/or EVs use protein channels or specific structures that guide them to the extracellular environment [59].

The wide variety of strains and diversity of environments for which vesiculation and the release of bacteria vesicles are important processes are quite remarkable and are also necessary, advantageous and common for all Gram-negative bacterial growth and survival [60]. Indeed, studies that have used vesiculation mutants have revealed that vesiculation is not a consequence of bacterial lysis or the disintegration of the bacterial envelope and cannot be simply correlated with membrane instability. The identification of a few low-vesiculation mutants and no null-vesiculation mutants suggests that this process may be important in Gram-negative bacteria growth [60]. Indeed, the main objective for bacteria in producing vesicles is their own survival, which allows the release of misfolded proteins or toxic material to remove surface-attacking agents, which contributes to nutrient acquisition. As for the advantages of vesiculation, this process allows the secretion of bacterial insoluble compounds such as adhesins for cellular aggregation in *Porphyromonas gingivalis* OMVs by mediating the coaggregation of different species in oral biofilms [61], which contributes to host interaction and colonization, the evasion of immune defense mechanisms and the destruction of host tissues. Vesicles are also a protective complex for bacterial compounds by allowing compounds to cover longer distances at high concentrations and can be specifically targeted to a specific destination such as receptors or ligands [62]. Currently, these vesicles are considered to be possible effective mechanisms for delivering drugs to exact tissues or cells [61].

### 5.2. Molecular Composition of Microbiota-Derived EVs

Recent studies have evidenced that the content in EVs differs between distinct bacterial species and even the same bacteria can produce different vesicles when external conditions change [63]. Bacterial vesicles contain diverse molecules such as proteins, lipids, nucleic acids and LPS. The identified bacterial EV molecules are listed in the EVpedia database (http://microvesicles.org/index.html, accessed on 19 April 2021) [64]. More than 100 vesicular proteins have been identified and most are associated with the extracellular region of Gram-negative bacteria such as the outer membrane, periplasm and inner membrane. The biological processes in which these proteins participate are the outer membrane assembly, pathogenesis, protein folding, protein insertion into the membrane and siderophore transport [65]. The virulence of bacterial EVs may differ between the bacteria of the same species. For instance, some *Vibrio cholera* strains, which, at a low cell density, produce OMVs with less peptidoglycan inside than at a high cell density, present a weaker and strong immunogenic effect, respectively [66]. These differences in composition may also bring about variations in the morphology of EVs [67].

One important factor to consider is that something positive for a parasite organism is negative for the host and vice versa. When the host reacts to an infection to eradicate it, the stressful environment can cause the production of bacterial EVs to increase. EVs contain virulence factors such as proteins (e.g., toxins, enzymes and adhesins) and non-protein antigens (e.g., LPS) and these EVs are able to deliver these virulence components to both prokaryotic and eukaryotic cells [68]. OMVs from *Pseudomonas aeruginosa* activate a pro-inflammatory response (e.g., IL-8) in lung epithelial cells [67], *Moraxella catarrhalis* secretes OMVs by carrying the ubiquitous surface protein UspA1/A2, which mitigates the mechanisms of serum resistance [69]. OMVs from *Porphyromonas gingivalis* help other pathogenic bacterial species to aggregate [70], *Mycobacterium tuberculosis* uses exosomes from its infected cells to transport lipoproteins and glycolipids to adjacent uninfected cells [71]. *Neisseria gonorrhoeae* can transfer plasmids to carry antibiotic resistance genes to other Gram-negative bacteria by using OMVs [72]. Bacteria in the gastrointestinal tract are no exception to these harmful effects of EVs. For instance, OMVs from *Helicobacter pylori* contribute to chronic gastritis and autoimmune diseases given the presence of LPS on their surface [73], protect bacterial cells against the toxic ROS produced by human polymorphonuclear cells [74] and increase IL-8 production in gastric epithelial cells [75]. *Campylobacter jejuni* OMVs improve the interaction and invasion between the microorganism and intestinal epithelial cells due to their proteolytic activity [76] while *Shigella flexneri* produces OMVs that can be fused with the outer membrane of other Gram-negative bacteria and their content is released in the recipient’s periplasm [77].

## 6. Activation of the Immune Response of TLRs by Microbiota-Derived EVs

TLRs are a family of innate immune system receptors and play key roles in protecting against microbial infections [78]. These receptors recognize the highly conserved motifs present in microorganisms, including bacteria, referred to as pathogen-associated molecular patterns (PAMPs) [79]. The PAMPs binding to TLRs initiate intracellular signaling pathways that lead to the up-regulation of a variety of costimulatory molecules and also to the synthesis and secretion of several cytokines and interferons by cells of the innate immune system. Ten TLRs have been identified in humans and twelve in mice [80] and they are divided into two groups based on their cell localization, cell surface or inside the cell that are able to recognize pathogen- or microorganism-associated molecular patterns (MAMPs) in extracellular or intracellular microbes, respectively [81]. TLRs 1, 2, 4, 5, 6 and 10 recognize MAMPs in bacterial wall components and flagellin from both Gram-positive and Gram-negative bacteria. For instance, TLR2 forms heterodimers with TLR1 or TLR6 and recognizes bacterial lipoproteins and lipopeptides such as peptidoglycans and lipoteichoic acid from Gram-positive bacteria. TLR4 recognizes LPS whereas TLR5 recognizes bacterial flagellin [81] (Figure 2).

The second group of TLRs (3, 7–9), which reside in intracellular compartments (e.g., endosomes and lysosomes), detects MAMPs in the nucleic acids deriving from bacterial and viral pathogens. TLRs 3, 7, 8 are able to recognize different types of RNAs from viruses but TLR9 recognizes the unmethylated CpG-containing DNA of a bacterial and viral origin [81] (Figure 2). Bacterial vesicles can also contain viral genetic material in the event of bacteria being infected by phages, which also involves a stressful condition that usually stimulates the release of EVs. Whereas access of cell surface TLRs to their extracellular ligands appears straightforward, the ligands for intracellular TLRs must be transported to cells and the mechanism of access of these ligands is poorly understood. However, the role of bacterial EVs is performed via this access, which has become more relevant in recent years.

Several reports have demonstrated the involvement of the activation of the immune response by OMV components. One of the most studied components is LPS, which is present on the OMV surface. OMVs from *Helicobacter pylori* increase the IL-8 production and generate a low degree of gastritis with no significant differences between *Helicobacter pylori* strains insofar as starting cytotoxicity, which suggests the presence of *Helicobacter pylori* LPS as a probable mechanism for this IL-8 up-regulation [75]. Conversely, bacterial EVs may have a modulatory effect on the immune system to protect bacteria from the host´s response. *Bacteroides fragilis* produces OMVs with a polysaccharide that interacts with dendritic cells and enhances regulatory T-cells and anti-inflammatory cytokine production [82]. Furthermore, the *Bacteroides vulgatus* strain mpk induces a tolerant semimature phenotype in BMDCs using OMVs [83] while EVs from the *Bifidobacterium longum* KACC 91563 strain induce the apoptosis of mast cells [84]. *Helicobacter pylori* OMVs are not only able to induce the release of both pro-inflammatory IL-6 and anti-inflammatory IL-10 but also induce T-cells apoptosis [85].

In addition to the activation of the immune response of the TLRs, bacterial EVs are able to activate other immune pathways such as cytosolic NLRs (Figure 2). The peptidoglycan of OMVs from several pathogenic (e.g., *Helicobacter pylori*) and commensal (e.g., *Escherichia coli*) bacterial species induces a cytosolic innate immune response mediated by NLR family member NOD1 [86,87]. *Salmonella typhimurium* OMVs activate dendritic cells in a TLR4-dependent and TLR4-independent manner through LPS and antigens are recognized by *Salmonella*-specific B- and T-cells [88]. OMVs from strains *Shigella dysenteriae* and *Escherichia coli* O104:H4 and O157 contain the Shiga toxin [89,90,91], a toxin that impairs the intestinal microvasculature and elicits cytokine and chemokine production by resident macrophages to induce immunopathology [92].

Finally, the release of bacterial EVs can also be influenced by antibiotics. For instance, Mitomycin C is an antibiotic that is able to induce in *Shigella dysenteriae* Shiga toxin production, whose presence in vesicles increases as does the number of vesicles released by the bacteria [89].

## 7. Involvement of Microbiota-Derived EVs in Neuropathogenesis

The gastrointestinal microbiota plays pivotal roles in neurodevelopmental processes and brain functions through the microbiota-gut-brain axis. The dysregulation of this axis by endogenous and exogenous factors such as aging and stress accelerates the occurrence of psychiatric disorders such as depression, anxiety, autism and Alzheimer’s disease [3,93]. For instance, treating patients with inflammatory bowel disease by an anti-inflammatory drug alleviates the brain function involved in visceral sensitivity and cognitive-affective biases [94,95], which suggests that the pathogenesis of psychiatric disorders is closely associated with the gut microbiota. Recent reports have demonstrated the involvement of bacterial EVs in the pathogenesis of neurodegenerative and behavioral diseases. Although the precise mechanisms of the penetration of bacterial EVs from circulation to the brain tissue have not been clearly elucidated, this section reviews how bacterial EVs can induce brain damage, behavioral impairments and mental disorders among others.

### 7.1. Bacterial EVs from the Gut to the Brain

Recent evidence demonstrates that bacterial EVs can directly or indirectly promote neural pathologies [50,96] but until EVs reach the brain, it is necessary to cross both intestinal and cerebral barriers. Although considerable discrepancies exist in the findings that determine the entry routes of EVs into host cells, bacterial EVs use both the paracellular and/or transcellular pathways to cross the intestinal barrier. They can employ four main endocytosis pathways on the transcellular route: micropinocytosis and endocytosis mediated by clathrin, endocytosis mediated by caveolin and clathrin- and caveolin-independent mechanisms (e.g., membrane fusion or lipid raft formation) [87,97]. Differences in the uptake routes between EVs from different species may well be explained by the fact that bacteria species display a high heterogeneity between them and the composition of EVs is adapted to direct vesicles toward a specific uptake route to thus allow them to undergo ideal processing in the host cell to facilitate infection [98]. For instance, *Salmonella enterica* OMVs are able to pass through the *Salmonella*-containing vacuole and be exocytosed by infected macrophages [99]. In other situations, vesicles can alter intestinal permeability such as *Campylobacter jejuni* OMVs that cleave E-cadherin and occludin [76] and facilitate the paracellular pathway. Bacterial EVs have the capacity to spread to the whole body in mice intraperitoneally injected with marked OMVs. Therefore, the microbiota vesicles that are able to cross the epithelial membrane would disseminate throughout the body under normal conditions [100].

On the BBB, it has been revealed that bacterial EVs can cross this barrier, which increases their permeability by inflammatory or infectious processes. For instance, there are reports of the presence of bacterial nucleic acids in postmortem brains of patients with Alzheimer’s disease. This genetic material could have arrived in vesicles by taking advantage of the weakened BBB in Alzheimer’s patients [101], which would lead to EVs or pathogens (e.g., bacteria) crossing the BBB to colonize the brain. Although a recent study evidenced that bacterial OMVs could directly cross the BBB [50], the question as to whether they can cross the BBB has not yet been resolved. A recent report demonstrated the transport of OMVs from the circulation to brain microglial cells through the meninges [102] and suggested that the delivery of bacterial EVs to the brain could be associated with infection at any place in the body, which could cause a fatal immune response in the brain [102]. Another recent report evidenced that bacterial EVs could cross the BBB through the vagus nerve. Following an oral gavage of OMVs or LPS from *Aggregatibacter actinomycetemcomitans*, fluorescein-conjugated EVs more strongly accumulated in the hippocampus than fluorescein-conjugated LPS and, after vagotomy, only the transport of fluorescein-conjugated EVs was blocked. These results suggest that vagotomy lowered the bacterial 16S rDNA levels in the hippocampus of mice orally administered with bacteria or their EVs [96]. 

How bacterial EVs can indirectly promote neuronal pathologies through their cargo is well-established. The presence of EVs in the CNS would probably induce an inflammatory response apart from the other possible side effects. However, as far as we know, the problem as to whether gut microbiota EVs can leak into the systemic circulation, cross the BBB and directly modify the brain still stands [50]. A recent report showed that OMVs from *Aggregatibacter actinomycetemcomitans*, a periodontal pathogen, could cross the mouse BBB and affect the brain immune response. Indeed it has been proposed that *Paenalcaligenes hominis* EVs can cause an impairment of the cognitive function by translocating to not only the brain through the vagus nerve but also to blood [96].

In addition, bacterial EVs may also use transport to the brain as intracellular agents during activated leukocytes trafficking to the brain. The brain is constantly surveyed by the trafficking of activated lymphocytes and macrophages, which provides a Trojan horse mechanism for microbial entry into the nervous system across the BBB. In fact this mechanism is a well-recognized route by which viruses infect the brain [103]. Bacterial components such as peptidoglycan have been detected in brain lesions from patients with multiple sclerosis that were heavily infiltrated with blood-derived leukocytes [104], which suggested that these immune cells could be a bacterial entry into the CNS of multiple sclerosis patients.

The importance of EVs in infection is well-demonstrated with the delivery of active virulence factors and immunomodulatory molecules as well as their defensive roles, which all serve to enhance pathogenesis [105]. Thus, insights into the mechanisms of bacterial EVs entering across intestinal and brain barriers may allow the design of inhibitors to prevent the delivery of their toxic cargo and attenuate infections without selecting them for antibiotic resistance [106].

### 7.2. Bacteria EV-Induced Brain Disorders

It is common knowledge that bacteria express many molecules that activate immune signaling cascades by engaging receptors (e.g., TLRs, NLRs) and lead to an immense capacity for influencing the brain function [107]. However, recent pieces of evidence have demonstrated that not only can cell bacteria components activate brain immune responses, which could lead to brain dysfunction, but bacterial EVs might also be implicated in these processes as they contain the same bacterial components (e.g., LPS, RNA, DNA, proteins). Indeed, OMVs are utilized by pathogens and commensal bacteria to manipulate the host immune response [108] and can even be used as a means of transporting DNA and RNA to host cells [109,110]. For instance, the intracardial administration of *Aggregatibacter actinomycetemcomitans* OMVs to mice is able to cross the BBB and the secreted extracellular RNA cargos derived from OMVs. These vesicles can be taken up by the host macrophage cells and sRNAs inside OMVs by activating TNF-α expression via the TLR8 and NF-κB signaling pathway. Increased TNF-α expression in the mouse brain induced by these OMVs may cause inflammatory diseases such as Alzheimer’s disease [50]. Similarly, OMVs secreted by the periodontopathogen *Aggregatibacter actinomycetemcomitans* can successfully deliver extracellular RNAs to brain monocyte/microglial cells to cause neuroinflammation associated with the up-regulation of IL-6 through NF-κB activation [102]. These effects could be associated with the activation of TLR8 as the OMV RNA of *Aggregatibacter actinomycetemcomitans* promotes the activation of the NF-κB response via TLR8 in macrophage U937 cells [50]. These findings provide a novel pathogenic mechanism in neuroinflammatory diseases.

The excessive generation of gastrointestinal bacterial products due to gut dysbiosis such as LPS and kynurenine causes gastrointestinal and systemic inflammation, which lead to inflammatory bowel disease and neuroinflammation [111,112]. However, the role of bacterial EVs and/or bacteria in the activation of the neuroimmune response and cognitive impairment is an unresolved question. A recent study showed that an oral gavage of LPS or EVs isolated from *Paenalcaligenes hominis* caused cognitive impairment in aged mice whereas an oral gavage of both led to severer cognitive impairment These effects are associated with hippocampal inflammation by inducing TLR4- and IL-1R-mediated NF-κB activation in innate immune cells such as microglial cells and successively suppress NF-κB-mediated BDNF expression in the hippocampus. *Paenalcaligenes hominis* proliferation in the intestine of elderly individuals and aged mice can accelerate the absorption of its LPS into the brain via blood and the translocation of its EVs into the brain. This scenario suggests that cognitive function impairment by *Paenalcaligenes hominis* may attribute to the combined effects of its EVs and LPS [96].

Although the brain has been classically considered a sterile organ with no microorganism under normal conditions, a few populations dominated by *Proteobacteria phylum* [103] and other populations that are commensal in the gut could induce diseases in the brain. For instance, *Escherichia coli* K1 can form part of the adult gut microbiota but can also provoke neonatal meningitis at a high mortality rate by invading brain microvascular endothelial cells to penetrate into the CNS [113]. The interaction of this bacteria with brain microvascular endothelial cells is mediated by a protein named OmpA, which is present in the outer membrane [113] and also abounds in *Escherichia coli* OMVs [114]. OmpA, by interacting with the receptor named gp96, a member of the HSP90 family, leads to phosphatidylinositol 3-kinase activation, which contributes to *Escherichia coli* K1 invasion of brain microvascular endothelial cells. These results suggest that EVs might also be involved in neonatal meningitis as *Escherichia coli* is able to cross the BBB without altering the integrity of the endothelial monolayer in vitro and BBB permeability in vivo [113].

Considering the importance of understanding the interaction between the gut microbiota and brain disease, a recent metagenomics study of bodily microbiota in an Alzheimer’s disease mouse model identified microbiota species from blood EVs whose composition altered in these animals. These results suggest that blood EVs are useful markers for characterizing the microbiota composition in Alzheimer’s disease or other neurodegenerative diseases [115].

## 8. Conclusions

Recent evidence has demonstrated the role of bacterial EVs from the gut during neurodevelopmental processes and brain dysfunctions by modulating the immune response. These vesicles are able to cross the BBB and interact with the immune receptors of the TLRs of glial cells, which leads to the production of cytokines and inflammatory mediators and alters the brain function (Figure 3). However, the degree of penetration of bacterial EVs into the brain tissue from either blood peripheral system circulation or the gastrointestinal tract remains an important question to answer to uncover the function of bacterial vesicles and their relationship with brain diseases.

## Figures and Tables

**Figure 1 ijms-22-04235-f001:**
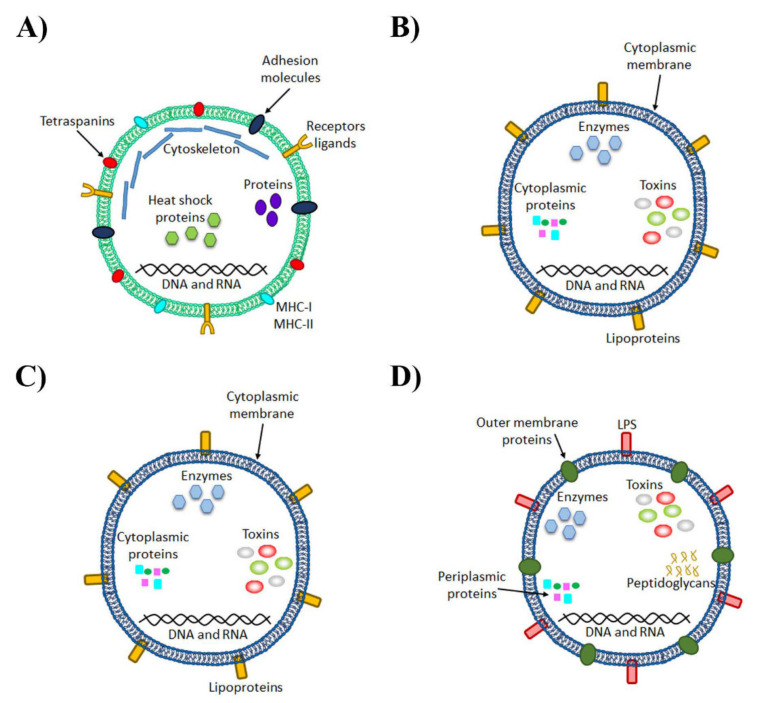
Architecture and composition of eukaryotic and bacterial extracellular vesicles. (**A**) Eukaryotic microvesicle, lipid bilayer-enclosed structures formed by the outward budding and fission of the plasma membrane with characteristic components such as flotillin-2, selectins, integrins, metalloproteinases and a high level of phosphatidylserine on the envelope. (**B**) Bacteria Gram-positive vesicles are comprised of the cytoplasmic membrane and also lipid bilayer-enclosed spheres and the cargo comes from the cytoplasm. (**C**) The outer membrane vesicles (OMVs) from Gram-negative bacteria are produced through outer membrane blebbing, whose cargo comes from the periplasm and contains peptidoglycan and periplasmic proteins and lipopolysaccharide (LPS) on their surface. (**D**) Outer-inner membrane vesicles (O-IMVs) are produced by Gram-negative bacteria under extreme stress or explosive cell lysis and contain a double bilayer showing inner membrane proteins and cytoplasmic proteins.

**Figure 2 ijms-22-04235-f002:**
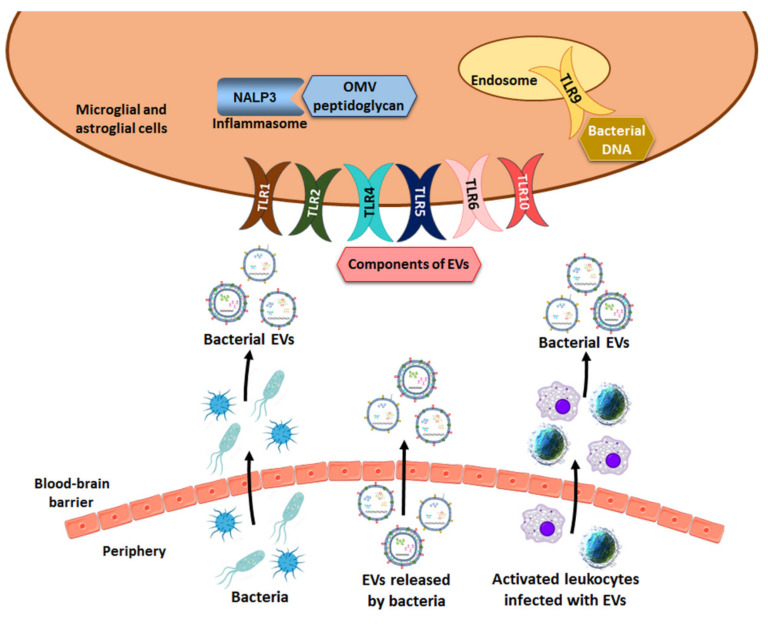
Scheme of the three possible mechanisms used by bacterial vesicles to penetrate the BBB. EVs can cross the BBB by themselves, in infected immune cells or through bacteria in different disorders. Once bacterial EVs are inside the brain, membrane components and the cargo of vesicles act as ligands of innate immune receptors (e.g., TLRs, NALP3 inflammasome) and activate the inflammatory immune response.

**Figure 3 ijms-22-04235-f003:**
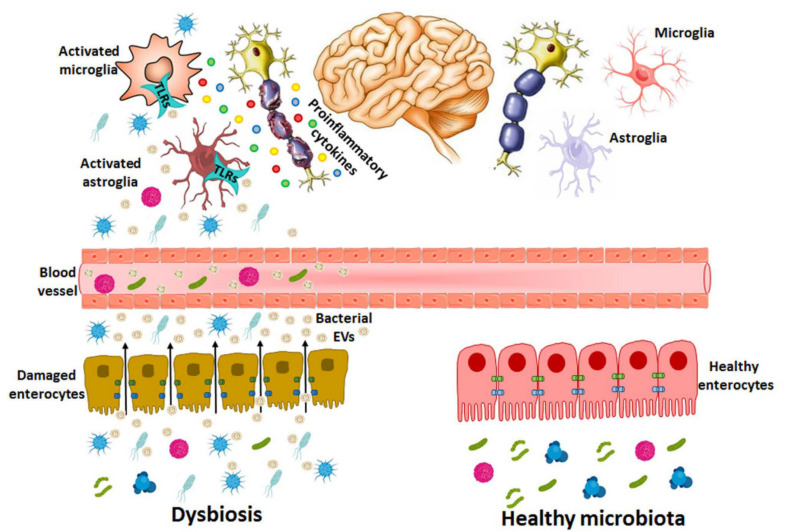
Bacterial EVs released during dysbiosis are able to cause brain disorders. In dysbiosis, the abundance of pathogenic bacterial species and harmful molecules such as LPS, peptidoglycans and toxins increase in the gastrointestinal tract and the intestinal epithelium is damaged by both bacterial activity and the inflammatory immune response, which increases the permeability and the transfer of EV components (e.g., LPS, RNA, DNA, proteins) from the intestinal lumen to the bloodstream. EVs might use paracellular and/or transcellular pathways to cross the intestinal barrier or be released by the bacteria already in blood vessels. If these vesicles reach the CNS, they activate immune cells (e.g., astrocytes and microglia) through the immune receptors of TLRs by triggering pro-inflammatory cytokines and causing neuronal damage.

## Data Availability

Not applicable.

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
