# Peer review of "Role of Microbiota-Derived Extracellular Vesicles in Gut-Brain Communication"

_ijms, 2021, doi:10.3390/ijms22084235_

Round 1
Reviewer 1 Report
In this review, authors discuss the role of microbiota-derived extracellular vesicles (EVs) in gut-brain communication, and their involvement in brain dysfunctions i.e. neuropathogenesis.
The topic is interesting, but description and organization of the content needs a considerable attention.
There are several issues that authors need to revise.
Major: since EVs mediate bidirectional communication, here in this cases from gut to brain and from brain to gut. Currently, the contribution of 'bacterial EVs' are discussed for neuropathogenesis but their contribution in the 'abnormalities of gut' are not covered. When it comes to role of bacterial EVs in the communication between gut and brain, authors should also discuss and list the abnormalities of gut, caused by bacterial EVs.
(1). The abstract is a very general summary, giving a superficial overview of the subject. I suggest authors to furnish the abstract with a bit mechanistic insight and also mention the name of key molecules (harmful bacterial EV content) involved in such communication and thus functional consequences.
(2). Abstract mentions the involvement of bacterial EVs in neuro developmental processes, but main body of the manuscript does not (or rarely) discuss the neuro ‘’developmental’’ processes.
(3). The introduction part needs more literature about the gut microbiota associated conditions
- Details about listed gastrointestinal and extra-gastrointestinal diseases.
- Also, the underlying aspects of bidirectional communication between the CNS and gut microbiota.
- Additionally, instead of mentioning the names of pathways provide the names of molecular players which contribute to these pathways, and how.
- Currently there are only two references in the introduction, which indeed needs more references to be included with recent updates.
(4). Page 1, line 35: The following statement appears that authors have performed the original study; ‘’We also analyze how bacterial EVs are capable of…..’’.
- Perhaps better to rephrase it, we also review how bacterial EVs are capable of..…
(5). Section 4: Role of extracellular vesicles (EVs) in the host: This part about EVs needs more literature. There are plenty of reviews already published, each review should provide new yet latest developments in the field. I suggest authors to include more studies about EVs in the host. Again, there is plenty of literature available.
(6). Section 7 and its subheadings. This part is superficial and does not provide functional insight (Keep in view that this is the central part of the review). I recommend authors to indicate the name of molecules transported by EVs, the state of donor/parent cells and of host or recipient cells. What regulations are driven by those molecules e.g., if EV transfers miRNA or TLRs to brain, what are their target molecules, pathways, and what kind of phenotypic and functional changes do occur. There are studies which have reported this. Please such provide description of the cited references instead of simply mentioning the general summary of a reference.
(7). There are two figures which are labeled as figure 1.
- In the conclusions, authors cite it as figure 3. But in the legends, it is labeled as figure 1 (it should be figure 3).
(8). Legends of last figure (Figure 3) should mention the content of EVs responsible for those functions. EVs, by themselves, are not independent functional entities, rather it is the molecular content carried by EVs, which induces the functional changes. Mention the name of EV content (molecules) responsible for communication and resulting functions.
(9). While authors mention the ‘‘harmful molecules’’… such as??
Overall, the review contains several errors or typos, which needs attention.
Few examples but not limited to;
- Page 1, line 15: which leads to → which lead to
- Page 2, line 50: The healthy human microbiota0 performs. This zero?
- Figure ‘numbers’ cited in the text do not match with corresponding figure.
Author Response
(1). The abstract is a very general summary, giving a superficial overview of the subject. I suggest authors to furnish the abstract with a bit mechanistic insight and also mention the name of key molecules (harmful bacterial EV content) involved in such communication and thus functional consequences.
As suggested, we have now incorporated some harmful bacterial EV content, such as lipopolysaccharide (LPS), peptidoglycans, toxins and nucleic acids that may have damaging consequences (see line 18).
(2). Abstract mentions the involvement of bacterial EVs in neuro developmental processes, but main body of the manuscript does not (or rarely) discuss the neuro ‘’developmental’’ processes.
Thank you very much for your comment. The term “neurodevelopment” has been eliminated.
(3). The introduction part needs more literature about the gut microbiota associated conditions
- Details about listed gastrointestinal and extra-gastrointestinal diseases.
- Also, the underlying aspects of bidirectional communication between the CNS and gut microbiota.
Additional information and reviews are now provided in the Introduction about gastrointestinal and extra-gastrointestinal diseases as well as bidirectional communication between the CNS and gut microbiota (p. 1).
- Additionally, instead of mentioning the names of pathways provide the names of molecular players which contribute to these pathways, and how.
- Currently there are only two references in the introduction, which indeed needs more references to be included with recent updates.
As suggested, we have incorporated several reviews in the introduction.
(4). Page 1, line 35: The following statement appears that authors have performed the original study; ‘’We also analyze how bacterial EVs are capable of…..’’.
- Perhaps better to rephrase it, we also review how bacterial EVs are capable of..…
Thank you very much for drawing our attention. We have corrected this and other typos.
(5). Section 4: Role of extracellular vesicles (EVs) in the host: This part about EVs needs more literature. There are plenty of reviews already published, each review should provide new yet latest developments in the field. I suggest authors to include more studies about EVs in the host. Again, there is plenty of literature available.
We appreciated your comment, and we have now included recent reviews on EVs in Section 4 (lines 136-137).
(6). Section 7 and its subheadings. This part is superficial and does not provide functional insight (Keep in view that this is the central part of the review). I recommend authors to indicate the name of molecules transported by EVs, the state of donor/parent cells and of host or recipient cells. What regulations are driven by those molecules e.g., if EV transfers miRNA or TLRs to brain, what are their target molecules, pathways, and what kind of phenotypic and functional changes do occur. There are studies which have reported this. Please such provide description of the cited references instead of simply mentioning the general summary of a reference.
We thank the reviewer for this relevant point. We have included in Section 7 a major description of the cited references, specifying in more detail the target molecules, pathways, functional changes, etc. (p. 10).
(7). There are two figures which are labeled as figure 1.
- In the conclusions, authors cite it as figure 3. But in the legends, it is labeled as figure 1 (it should be figure 3).
This typo has been corrected.
(8). Legends of last figure (Figure 3) should mention the content of EVs responsible for those functions. EVs, by themselves, are not independent functional entities, rather it is the molecular content carried by EVs, which induces the functional changes. Mention the name of EV content (molecules) responsible for communication and resulting functions.
As suggested, we have incorporated in the legend of Figure 3 several harmful molecules (e.g., LPS, peptidoglycans, toxins), which are present in EVs and are responsible for communication and resulting functions (p. 11).
(9). While authors mention the ‘‘harmful molecules’’… such as??
Overall, the review contains several errors or typos, which needs attention.
Few examples but not limited to;
Page 1, line 15: which leads to → which lead to
Page 2, line 50: The healthy human microbiota0 performs. This zero?
Figure ‘numbers’ cited in the text do not match with corresponding figure.
These typos have also been corrected.
Reviewer 2 Report
The article is timely and well-written.
Minor suggestions: More thorough literature review would make the article more readable. Below are some examples:
1. In the first paragraph, "Notably dysfunctions or alterations in microbial imbalance populations, named dysbiosis, are associated with different diseases, including mental disorders, brain dysfunction or behavioral impairments [1]." In addition, other brain disorders such as Alzheimer disease are also included. https://pubmed.ncbi.nlm.nih.gov/32687964/ Though the authors mention Alzheimer later in the article, they can also address it earlier.
2. The paper illustrated several of the main pathways of the potential communication between the gut microbiota and the brain. In addition, other pathways such as neurotransmitters may also be involved. https://pubmed.ncbi.nlm.nih.gov/32475
Author Response
1. In the first paragraph, "Notably dysfunctions or alterations in microbial imbalance populations, named dysbiosis, are associated with different diseases, including mental disorders, brain dysfunction or behavioral impairments [1]." In addition, other brain disorders such as Alzheimer disease are also included. https://pubmed.ncbi.nlm.nih.gov/32687964/ Though the authors mention Alzheimer later in the article, they can also address it earlier.
Thank you very much for your suggestion, which is now incorporated.
2. The paper illustrated several of the main pathways of the potential communication between the gut microbiota and the brain. In addition, other pathways such as neurotransmitters may also be involved. https://pubmed.ncbi.nlm.nih.gov/32475
As suggested, we have introduced the neurotransmitters as other pathway involved in gut-brain communication (lines 104-107).
Round 2
Reviewer 1 Report
The authors have addressed the points and have further expanded the level of discussion in this review.
I endorse this publication with a minor comment that there are certain typos that may need attention during the production of this article.
Good luck!